# Cytomegalovirus Infections in Children with Primary and Secondary Immune Deficiencies

**DOI:** 10.3390/v13102001

**Published:** 2021-10-05

**Authors:** Caroline M. Bateman, Alison Kesson, Madeleine Powys, Melanie Wong, Emily Blyth

**Affiliations:** 1Cancer Centre for Children, Children’s Hospital at Westmead, Westmead, Sydney, NSW 2145, Australia; caroline.bateman@sydney.edu.au (C.M.B.); madeleine.powys@health.nsw.gov.au (M.P.); 2Westmead Institute for Medical Research, University of Sydney, Westmead, Sydney, NSW 2145, Australia; 3Department of Infectious Diseases and Microbiology, Children’s Hospital at Westmead, Westmead, Sydney, NSW 2145, Australia; alison.kesson@health.nsw.gov.au; 4Discipline of Child and Adolescent Health, Sydney Medical School, Faculty of Medicine and Health, University of Sydney, Sydney, NSW 2145, Australia; 5The Marie Bashir Institute for Infectious Diseases and Biosecurity, University of Sydney, Sydney, NSW 2145, Australia; 6Department of Allergy and Immunology, Children’s Hospital at Westmead, Sydney, NSW 2145, Australia; melanie.wong@health.nsw.gov.au; 7Blood Transplant and Cellular Therapies Program, Department of Haematology, Westmead Hospital, Westmead, Sydney, NSW 2145, Australia; 8Sydney Medical School, Faculty of Medicine and Health, University of Sydney, Sydney, NSW 2145, Australia

**Keywords:** child, cytomegalovirus, haematopoietic stem cell transplant

## Abstract

Cytomegalovirus (CMV) is a human herpes virus that causes significant morbidity and mortality in immunosuppressed children. CMV primary infection causes a clinically mild disease in healthy children, usually in early childhood; the virus then utilises several mechanisms to establish host latency, which allows for periodic reactivation, particularly when the host is immunocompromised. It is this reactivation that is responsible for the significant morbidity and mortality in immunocompromised children. We review CMV infection in the primary immunodeficient host, including early identification of these infants by newborn screening to allow for CMV infection prevention strategies. Furthermore, clinical CMV is discussed in the context of children treated with secondary immunodeficiency, particularly paediatric cancer patients and children undergoing haematopoietic stem cell transplant (HSCT). Treatments for CMV are highlighted and include CMV immunotherapy.

## 1. Introduction

Human cytomegalovirus (CMV) or human herpes virus-5 is a member of the *Betaherpesvirinae* subfamily of the family *Herpesviridae* [1]. It is a double-stranded DNA virus that causes primary infection, usually in childhood; it is not cleared from the host and becomes latent in white blood cells. In immunocompetent children, primary CMV infection commonly causes a mild illness and is associated with lymphopenia, lymphadenopathy, fever and hepatosplenomegaly. In contrast, CMV has significant implications for children who are or become immunodeficient. This includes those with a primary immune disorder or a secondary immune disorder, acquired due to medical treatment such as immunosuppressive therapy, haemopoietic bone marrow transplant (HSCT) or solid organ transplant [2]. 

In this review, we discuss the clinical significance and standard and novel treatments of CMV in the primary immunodeficient host, those receiving immunosuppressive therapies for cancer and post-allogeneic HSCT. 

## 2. Clinical CMV

CMV is a double-stranded DNA virus comprising approximately 235,500 base pairs encoding approximately 165 open reading frames. The genome comprises two unique regions, unique long and unique short, flanked by repeated sequences [3,4,5]. Following the initial infection, CMV is not cleared from the host and establishes lifelong latent infection in undifferentiated CD34^+^ stem cells and CD33^+^ myeloid progenitor cells and the CD14^+^ monocytes and dendritic cells that they mature into [6,7]. CMV is also latent in other tissues, such as lung [8]. How CMV is able to establish this lifelong latency is not entirely clear but involves a variety of mechanisms allowing it to evade the host innate immune response [9,10,11]. CMV has evolved sophisticated strategies to circumvent immune cell recognition, encoding arsenals of immunomodulatory molecules—termed immunoevasins—that seek to subvert T cell and natural killer function, allowing it to establish lifelong infections [12]. 

Restriction enzyme analysis of the CMV DNA demonstrate many genetic variants or strains; however, the differing strains do not allow for classification into distinct genotypes. Furthermore, the corresponding antigenic differences do not define differing serotypes. An individual who has been infected with one strain of CMV does not necessarily have protection from other CMV strains [13,14]. 

CMV infection can be life-threatening in immunocompromised patients. The broad cellular tropism of CMV results in a diverse range of pathologies and disease manifestations associated with infection. The infection can be a primary infection, a re-infection with a different strain of CMV or reactivation of the virus from latency. The immunosuppressed patients at risk are those with primary T cell immunodeficiency, HIV/AIDS patients, solid organ transplant patients and those undergoing chemotherapy for haematological malignancies with or without the additional immunosuppression, from syngeneic and allogeneic haemopoietic cell transplant as well as the foetus, leading to congenital CMV and neonates with perinatal CMV [2]. 

### 2.1. CMV in the Primary Immunodeficient Host 

Although CMV infection in healthy children and adults is usually mild or asymptomatic, immunocompromised individuals are at risk of more severe disease. Primary immunodeficiencies are a diverse group of disorders that affect 5.6 people per 100,000 in Australia. The most common primary immunodeficiency is antibody deficiency syndrome, which affects 77% of patients [15]. 

CMV can be transmitted to the neonate by several routes, including transplacentally, through maternal genital secretions during delivery and postnatally via maternal oral secretions, breast milk, objects contaminated with body fluids (e.g., utensils such as drink bottles, dummies/soothers) and via blood products. Local CMV reactivation occurs in the mammary glands at the beginning of lactation, and CMV DNA can be detected in the breast milk of 96% of CMV IgG positive mothers [16,17].

Neonates who are born at term and immunocompetent do not usually have significant CMV disease due to the presence of maternal antibody [18] but may develop mild neuro-developmental sequelae, most commonly neuro-sensory hearing loss. However, the premature neonate, very low birthweight neonate and the neonate with primary immunodeficiency are at a significant risk of severe CMV infection, including pneumonia, hepatitis, neutropenia, thrombocytopenia and enterocolitis [19,20]. Impaired innate and adaptive immune responses contribute to the disease severity [21]. 

Severe combined immunodeficiency (SCID), one of the most severe primary immunodeficiencies, is an inherited genetic disorder (most commonly X-linked) that leads to a combined disorder of T cells and B cells. Less than 10 children are born with SCID in Australia per year [22]. These infants usually die within the first 2 years of life due to infection unless diagnosed early, enabling restoration of a functioning immune system by haematopoietic stem cell transplantation. Affected babies appear normal at birth, with the first presentation often with life-threatening opportunistic infection, typically *pneumocystis jirovecii* pneumonia. Diagnosis in the asymptomatic stage enables commencement of prophylactic therapies and procedures to prevent opportunistic infection, irreversible organ damage and death, resulting in significantly improved morbidity and mortality.

The fundamental defect in SCID is the inability to produce naive T cells, with resultant severe T cell lymphopenia. Development of a robust test to detect T cell receptor excision circles (TRECs), as a marker of naive T cell production, has enabled the establishment of a SCID newborn screening test in many countries, mostly with modern healthcare systems. Other causes of significant T cell lymphopenia are also detected [23]. Positive TREC newborn screening must be confirmed by immunoglobulin levels, lymphocyte subsets, mitogen-induced lymphocyte proliferation and genetic testing. The outcome for infants identified early and transplanted before the age of 3.5 months is excellent (95–100% survival), with slightly lower survival if transplanted after this time but without acquisition of infection (~90% survival), with significant impact seen if transplanted after the age of 3.5 months, having acquired an infection that was resolved by the time of transplant (~80% survival) and even worse if transplanted in the setting of active infection, CMV being predominant (50–60% survival) [24]. The shortened time interval from birth to intervention may prevent CMV transmission [25].

Strategies to reduce the risk of transmission include withholding of breastfeeding whilst maternal CMV status is determined, with cessation if positive, use of CMV negative and irradiated blood products infused with a leucocyte filter and avoiding close contact with young children.

As seen in AIDS due to HIV infection, individuals with T cell dysfunction in the setting of combined immunodeficiencies are also at risk of severe CMV infection. These immunodeficiencies can present in infants, older children and young adults with a range of features, including Omenn’s syndrome, autoimmunity, granulomas, as well as predisposition to infections. In these individuals, persistent CMV can also drive progression to lymphoid malignancy [21]. A regularly updated comprehensive list of recognised immunodeficiencies is published by the International Union of Immunological Societies (IUIS) [26]. 

In some primary immunodeficiencies, susceptibility to infection with gamma-herpes viruses such as CMV and Epstein–Barr virus (EBV) dominates the clinical picture [26]. These conditions usually involve cytolytic T cell defects, with or without concomitant NK cell defects. In these disorders, in affected individuals, CMV infection can lead to haemophagocytic lymphohistiocytosis (HLH). HLH is a life-threatening condition characterised by excessive immune activation and can be diagnosed either by a molecular diagnosis consistent with familial HLH or clinically when patients meet five out of eight criteria: fever, splenomegaly, cytopenias affecting two or more blood lineages, hypertriglyceridemia and/or hypofibrinogenemia, haemophagocytosis, low/absent NK cell activity, hyperferritinemia and high soluble interleukin-2 receptor levels [27]. Early diagnosis and aggressive treatment of HLH is essential to avoid morbidity and mortality. HSCT in remission, the only curative therapy, is the standard of care to prevent life-threatening relapse when an underlying genetic cause is found [28]. 

In addition to the induction of HLH in individuals with HLH-associated genetic defects, active and latent CMV infection induces sustained systemic inflammatory responses and immune dysregulation and predisposes patients to the development of autoimmune phenomena [29]. CMV also causes further immunosuppression associated with T cell exhaustion, contributing to the persistence of infection [30].

At the other end of the age spectrum, inherited immune deficiency in adults can be responsible for rare cases of significant CMV disease, as demonstrated by an isolated case of fatal disseminated CMV infection in a previously well 51-year-old adult with autosomal recessive NOS2 deficiency [31].

### 2.2. CMV in the Paediatric Cancer Patient 

In the last 50 years, successive international clinical trials have allowed for the improved survival of children diagnosed with cancer, especially childhood leukaemia [32]. This improvement is attributable to the incremental effect of new anti-cancer agents and their combinations, better supportive care including intensive care, broader-spectrum antibiotics/antivirals and the availability of blood products [33]. Although this has clearly been a triumph of clinical medicine, there are still too many children either dying of cancer or having long-term side effects due to their treatments. 

#### 2.2.1. Cytotoxic Chemotherapy

For many paediatric cancers, the backbone of curative treatment is with cytotoxic chemotherapy. These agents have limited tolerability as they cause significant mucocutaneus inflammation, myelosuppression and immunosuppression and other longer-term side effects. In the search for a cure for all, the intensity of treatment using cytotoxic drugs has been escalated, particularly for those patients at higher risk of relapse; with this intensity escalation, the toxicities have also mounted. This led to the development of novel agents that were efficacious against cancer, having a side effect profile different from conventional cytotoxic medications [34]. These more targeted agents, such as the Janus kinase (JAK2) inhibitor Ruxolitinib and Bruton’s kinase inhibitor Ibrutinib, although not widely used in children, are associated with an increased risk of viral infections and can be additive to other agents in terms of CMV risk.

Some of the cytotoxic chemotherapy agents are known to be preferentially lymphocyte depleting; these include, but are not limited to, the purine analogues, such as fludarabine, clofarabine and nelarabine. These agents have demonstrated efficacy in both adult and paediatric malignancies but are also cytotoxic to T cells [35], placing the recipients at increased risk of CMV infection [36]. Fludarabine is becoming increasingly popular due to this lymphodepleting effect, for use prior to adoptive T cell transfer to facilitate their expansion and persistence [37]. It is likely that the use of fludarabine contributes to CMV risk in this group of patients. 

#### 2.2.2. Immunotherapy—Pharmaceutical Agents

Immunotherapies are a diverse group of agents that act primarily through engagement with the immune system for tumour control. This includes monoclonal antibodies, antibody drug conjugates (ADC), bi- or tri-specific engager molecules and immune checkpoint inhibitors. These agents are of relevance to infection risk when they specifically interact with T or B cells and induce tumour death by immune interactions. This potentially will have an impact on CMV in cancer patients receiving these agents.

The most established immunotherapeutic drugs used in the treatment of cancer is Rituximab. Rituximab is a monoclonal antibody with its epitope against the cell antigen CD20 [34]. CD20 is commonly found on mature B cells, leading to its therapeutic use in B cell neoplasms, EBV disease and some autoimmune disorders. The mechanism of action of Rituximab is known to cause prolonged B cell depletion by its engagement with CD20 on normal B cells; however, the extent of this immune destruction and the association with CMV are poorly identified and reported. As Rituximab is now incorporated into many B cell malignancy treatment regimens, with proven survival advantages in both children [38] and adults [39], and given in combination with other T cell depleting agents, it can be difficult to determine the effect that Rituximab is having on the T cell repertoire [40]. There are several case series of CMV disease in a diverse group of patients, such as those with immunological disorders [41], but the lack of robust data is striking in adults, and even more so in children. 

There are substantial numbers of antibody drug conjugates used for the treatment of haematological malignancies in children. Gemutuzumab ozogamicin (GO) is used in the treatment of acute myeloid leukaemia (AML) in adults and children [42]. GO is a humanised anti-CD33 antibody conjugated to a calicheamicin derivative that is stable in the circulation but, once internalised, the calicheamicin toxin is released, leading to DNA binding and cell death [43]. The absence of specific reports of GO associated with CMV disease are most likely due to the difficulty of separating GO from the immunological effects of chemotherapy copartners, as there are many reports of CMV disease in AML patients. Of emerging interest with GO is its interaction with myeloid-derived suppressor cells (MDSCs). GO has the potential to deplete MDSCs, changing the immunosuppressive microenvironment of tumours to reactivate T cell immunotherapy or restoring T cell immunity in the immunodeficient [44]. The antibody drug conjugates are a diverse group, and the T cell interactions are not fully characterised. This is a group of drugs that is still being explored, with recent publications on anti-CD7 ADC showing promising potent and selective effects against CD7-expressing cells in preclinical data [45]. This agent has a potential on-target off-tumour impact on CD7-expressing normal T cells.

#### 2.2.3. Cellular Immunotherapies

Chimeric antigen receptor T cells (CAR T) with CD19-targeted chimeric antigen receptor constructs are now the standard of care for relapsed or refractory B-ALL in children and young adults, with a large number of cellular cancer therapies in the development pipeline for B-ALL and other malignant diseases. The success of the first group of FDA-approved CAR T products relies on their action against CD19, a pan B cell antigen. Ablation of the B cell compartment leads to hypogammaglobulinaemia and nearly all patients require immunoglobulin replacement. Viral infection risk in CAR T recipients is reported at around 9% of all CAR T recipients, with emerging data suggesting that it is usually early post-CAR-T infusion (less than 90 days) and associated with those who receive a higher CAR T dose [46].

### 2.3. CMV in the HSCT Recipient

CMV infection and disease is common following allogeneic HSCT and, despite advances in diagnosis and pre-emptive therapy, still causes significant morbidity and mortality [47]. CMV viraemia occurs in approximately 25% of paediatric allogeneic HSCT recipients post-transplant, occurring mostly early post-transplant [47,48]. This is a lower incidence than in adults due to the higher rate of seronegativity in the paediatric recipient and donor population. Some patients with post-transplant CMV viraemia will develop CMV-related organ disease, the most significant manifestations of which are pneumonitis, hepatitis, gastroenteritis, retinitis and encephalitis [48]. CMV viraemia and disease post-transplant continue to be associated with increased non-relapse mortality (NRM) despite improvements in antiviral therapy and the understanding of CMV T cell reconstitution [49]. CMV infection post-transplant is also associated with increased risk of acute and chronic graft versus host disease (GVHD), secondary bacterial and fungal infections, prolonged hospital admission and substantially increased treatment costs [50]. Current prophylactic and pre-emptive strategies used to manage CMV reactivation in paediatric allogeneic HSCT recipients effectively prevent CMV disease but are associated with significant side effects, and the optimum approach remains unclear [51]. New anti-CMV drugs such as letermovir and adoptive CMV-specific T cell therapies offer promise, and it is hoped that future CMV prevention and management strategies will be less toxic and able to be tailored to a patient’s individual risk of CMV disease [52]. 

The risk of CMV reactivation and disease post-transplant is dynamic over time, and depends on multiple factors, including the CMV serostatus of donor and recipient, donor type, stem cell source, T cell depletion of the graft, conditioning regimen and GHVD [52] Patients are at risk of CMV reactivation if the donor and/or recipient are CMV seropositive pre-transplant and the risk is predictable based on the presence of donor-derived antigen-experienced immune effectors that can control largely recipient-derived reactivation episodes. With modern blood banking leukodepletion processes and judicious use of CMV negative red cell and granulocyte transfusion, the rate of transfusion-associated primary CMV is rare. Studies of paediatric and adult patients post-HSCT report the incidence of CMV viraemia post-transplant as 40 to 70%, although there is variability in definitions of viraemia and diagnostic methods [53]. Recipients at high risk of CMV reactivation include any of the following [52]:(a)family donor with >1 mismatch at HLA-A, B, C, including haploidentical donor;(b)unrelated donor with >1 mismatch at HLA-A, B, C, DRB1;(c)cord blood transplant or PBSC graft [50];(d)T cell depletion of the graft: (a) ex vivo with antithymocyte globulin (ATG) or alemtuzumab (humanised IgG1 anti-CD52); (b) in vivo with techniques such as CD34 + selection;(e)Grade > 2 acute GVHD requiring >1 mg/kg/day of prednisolone.

Overall, 98% of CMV reactivation occurs prior to day +100, with studies reporting a median time to onset of 20–71 days [47]. Early CMV reactivation is associated with reduced CD4 T cell recovery (CD 4 < 0.15 × 10^9^/L), which can be due to T cell depletion of the graft or GVHD requiring steroids [50]. Early CMV reactivation is associated with reduced overall survival and does not confer a reduced risk of relapse for haematological malignancies [52,54].

The pretransplant CMV serology of the recipient and donor is a key factor in allogeneic donor selection, secondary only in importance to the HLA match [48]. Relative risk of reactivation can be stratified as follows—“R+/D− > R+/D+ > R−/D+”; there is almost zero risk with R-/D- as long as primary infection via transmission in the community does not occur. Concordance of the donor and recipient’s CMV serostatus is preferred where possible, noting that cord blood is treated as CMV seronegative. Although the incidence of CMV seropositivity increases with increasing age, most paediatric HSCT recipients are CMV seropositive pre-transplant [47]. Recipient CMV seropositivity is associated with decreased overall survival and CMV seropositive donors are preferred for CMV seropositive recipients to establish earlier CMV-specific immune-reconstitution post-HSCT [55]. For CMV seronegative recipients, a CMV negative donor is a preferred over a CMV seropositive donor whenever possible, even if there are mismatches at HLA-C, -DQ, -DP, as CMV seropositive grafts are associated with decreased overall survival (OS) in this setting [55]. Furthermore, the risk of transmission of CMV by the stem cell product to the recipient in the case of a CMV positive donor and CMV negative recipient is 20 to 30% [55]. For CMV seronegative recipients, the CMV serostatus of a matched sibling donor does not appear to affect OS [55]. CMV seronegative recipients should also receive CMV seronegative, leukocyte-reduced, filtered blood products to reduce the risk of CMV transmission [34]. 

## 3. CMV Therapeutic Strategies 

### 3.1. Antiviral Pharmacotherapy

#### 3.1.1. Ganciclovir

The most frequently used drug for CMV disease in the immunocompromised population, including children, is intravenous ganciclovir (GCV), a synthetic nucleoside guanine analogue. GCV is given intravenously as it has very poor oral bioavailability. The main site of action of GCV is the DNA polymerase, UL54. Ganciclovir is a prodrug that must first be phosphorylated for activity. The initial mono-phosphorylation is mediated by the CMV *UL97* encoded kinase, resulting in ganciclovir’s selectivity for infected rather than uninfected cells [56]. This is followed by further phosphorylation mediated by cellular kinases to a tri-phosphate form. When ganciclovir is converted to its tri-phosphate form, it inhibits the viral DNA polymerase, UL54, resulting in blocking DNA chain elongation [56]. *UL97* mutations arise more frequently than *UL54* mutations. Virus-containing mutations in the *UL97* gene are resistant to ganciclovir alone, with M460V/I, H520Q, C592G, A594V, L595S, and C603W being the most frequently reported ganciclovir-resistance-associated amino acid substitutions [57]. A wide variety of *UL54* mutations can arise in drug-resistant CMV after prolonged exposure to ganciclovir and can add to the phenotypic resistance already conferred by *UL97* mutations [57]. In contrast to mutations in *UL97*, mutations in *UL54* often lead to less fit virus with retardation of viral growth in cell culture, implying that *UL54* mutations may not be the major genetic pathway of drug resistance [58]. 

Ganciclovir has significant cellular toxicity, the main adverse events being neutropenia, anaemia, thrombocytopenia, diarrhoea and fever. The ganciclovir-induced blood dyscrasias are of the most concern in patients undergoing haemopoietic stem cell therapy with neutropenia, increasing the risk of bacterial and fungal infection. Ganciclovir is used for the treatment of sight-threatening CMV disease in AIDS and other severely immunocompromised patients and confirmed CMV pneumonitis in bone marrow transplant patients. Prophylaxis of CMV in solid organ transplant and marrow transplant patients with ganciclovir can be given, generally for defined periods, usually not more than 100 days to limit drug-induced neutropenia and nephrotoxicity. However, CMV disease may still occur after this period as antiviral drugs block viral replication but do not eradicate the virus and, additionally, ganciclovir has no action against latent CMV. Neutropenia is often dose-limiting, often early in therapy, and can be reversed by ganciclovir discontinuation. The ganciclovir-induced neutropenia can preclude the daily use of ganciclovir for CMV prophylaxis after haemopoietic stem cell transplant. In these patients, the CMV viraemia or viral load is monitored with pre-emptive therapy if the viral load exceeds a pre-defined threshold.

Toxicity may be enhanced when ganciclovir is co-administered with other myelosuppressive or nephrotoxic medications. There are animal studies showing that ganciclovir can reduce fertility and spermatogenesis. Safety in pregnancy has not been established as ganciclovir may have mutagenic and teratogenic potential. 

#### 3.1.2. Valganciclovir

Valganciclovir is a prodrug of ganciclovir, which, unlike ganciclovir, is well absorbed from the gastrointestinal tract in children and rapidly metabolised to ganciclovir in the intestinal wall and liver. The absolute bioavailability of ganciclovir from valganciclovir is approximately 60%. The toxicity and resistance mutations of valganciclovir are the same as ganciclovir. The main advantage of this agent is the ability to dose valganciclovir orally, with better absorption if administered with food. 

#### 3.1.3. Foscarnet

Foscaranet is a phosphonic acid derivative antiviral that preferentially inhibits the viral UL54 DNA polymerase compared with cellular polymerases. It has a broad antiviral spectrum that inhibits all known human viruses of the herpes group, herpes simplex 1 and 2, human herpes virus 6, varicella zoster virus, Epstein–Barr virus and CMV, at concentrations not affecting normal cell growth. Foscarnet also inhibits the viral DNA polymerases from hepatitis B virus. Foscarnet does not require intracellular activation by kinases and therefore is active in vitro against CMV *UL97* mutants without mutation in the *UL54* gene [57]. 

Foscarnet can impair renal function and should be used with caution in children with reduced renal function. Since renal function impairment may occur at any time during foscarnet administration, serum creatinine should be monitored at least every other day during induction therapy and weekly during maintenance therapy. Appropriate dose adjustments should be made if renal function is affected. Due to excretion in the urine, foscarnet can cause genital ulceration, particularly if contact with skin or mucosal surfaces is prolonged.

Foscarnet chelates bivalent metal ions, such as calcium, and foscarnet can cause an acute decrease in ionised serum calcium, not reflected in total serum calcium levels, which is proportional to the rate of infusion. Patients’ electrolytes, especially calcium and magnesium, should be assessed prior to and during foscarnet therapy and deficiencies corrected. Foscarnet can also cause cardiac arrhythmias. 

Renal toxicity may increase when foscarnet is used in combination with other nephrotoxic drugs such as aminoglycosides, amphotericin B, ciclosporin A, methotrexate and tacrolimus [27]. While foscarnet is an effective anti-CMV agent, these many challenges with administration mean that it is reserved for second or third line if ganciclovir or valganciclovir have failed or are contraindicated.

#### 3.1.4. Cidofovir 

Cidofovir is a cyclic nucleoside phosphonate analogue of cytosine that is active against CMV by competitively inhibiting the incorporation of deoxycytidine triphosphate into viral DNA by viral DNA polymerase, resulting in disruption of DNA chain elongation [59]. Cidofovir is not a potent anti-CMV agent and its use in children is limited by this as well as its side effect profile. There is a significant incidence of nephrotoxicity. Proteinuria is an early and sensitive indicator of cidofovir-induced nephrotoxicity. Oral probenecid is usually given with each cidofovir dose to reduce nephrotoxicity; however, probenecid is known to interact with the metabolism or renal tubular secretion of many drugs [27]. 

Cidofovir can cause neutropenia, and myelosuppression should be monitored during and after treatment.

#### 3.1.5. Brincidofovir

Brincidofovir is a novel prodrug of cidofovir that can be taken orally. It is conjugated to a lipid, resulting in increased tissue distribution and intracellular release with a prolonged half-life, and is approximately 100-fold more potent than cidofovir. Brincidofovir has enhanced antiviral action compared with cidofovir, with reduced renal toxicity [60]. However, the results of a randomised, double-blind, placebo-controlled trial of brincidofovir as prophylaxis in HSCT were disappointing: although this study demonstrated reduced invasive CMV infection up to week 14 post-HSCT, there were more serious adverse events in the brincidofovir arm, with all-cause mortality being higher in the brincidofovir arm compared to placebo by week 24 post-HSCT [61]. Further studies are needed due the possibility of investigator misinterpretation of diarrhoea as GVHD in this study. 

#### 3.1.6. Letermovir

Letermovir is a quinazoline CMV DNA terminase complex inhibitor. The DNA chain terminator is unique to herpes viruses and does not have a mammalian homologue. A trimer of proteins UL51/UL56/UL89 forms the terminase complex, which binds the CMV genome to the capsid. Resistance to letermovir has been shown to be due to mutations in *UL56* and, less frequently, *UL51* and *UL89*, the gene complex encoding the viral terminase. Thus, there is no cross-resistance between GCV, foscarnet and cidofovir with letermovir. Letermovir’s target is the UL56 of the CMV terminase trimer and it has been shown in cell culture to be significantly more potent than GCV in inhibiting CMV replication in vitro. Additionally, letermovir does not exert any antagonism on GCV, foscarnet or cidofovir and could possibly be used in combination; however, there are no data yet to support this dual mode of therapy. Letermovir, which is excreted by the liver, has altered pharmacokinetics in patients with kidney and hepatic failure with increasing levels and thus its use in kidney and liver transplant patients may be contraindicated [62]. Letermovir interacts with the immunosuppressants ciclosporin A and tacrolimus, increasing their exposure, with the dose of letermovir needing to be halved when co-administered with these drugs. 

A review of randomised, placebo-controlled trials of the efficacy of letermovir as prophylaxis for CMV disease showed significantly reduced reactivation of CMV and reduced mortality, especially in the high-risk patients undergoing haploidentical, mis-matched donor or T-cell-depleted grafts [52,63]. Recent studies in adult patients have demonstrated that when letermovir was used prophylactically or pre-emptively, it significantly reduced the incidence of CMV morbidity and mortality when compared with a placebo or historical controls [52,64,65]. 

Letermovir has a different toxicity profile to GCV, foscarnet and cidofovir and a different mechanism of action, resulting in different genetic mutations causing resistance. These variables need consideration when prescribing for CMV prophylaxis or therapy. Blinded non-inferiority studies need to be undertaken with letermovir and GCV to fully determine the relative efficacy, toxicity and cost of the various options for CMV therapies. 

#### 3.1.7. Maribavir (MBV)

Marabavir is a benzimidazole riboside and is an active oral antiviral directed against the CMV UL97, resulting in blocking the nuclear egress of viral capsids through the inhibition of UL97 [66]. Marabavir has a good safety profile [66], without associated myelosuppression or nephrotoxicity; however, co-administration of MBV and GCV is not advised as MBV inhibits the action of UL97, which is essential for initiating the phosphorylation activation cascade for GCV’s anti-CMV activity. MBV is still an investigational agent but may have a role in treating patients with GCV and cidofovir and foscarnet resistance [67,68]. 

#### 3.1.8. CMV Immunoglobulin

Human CMV immunoglobulin can be used for CMV infection prophylaxis following bone marrow or renal transplantation and as an adjunct for the treatment of CMV infections such as pneumonitis. It may be of use in patients with poor tolerance of CMV antiviral drugs due to toxicity or those patients with hypogammaglobulinaemia. The role of CMV immunoglobulin remains unclear; at best, it has a limited role in the prevention or treatment of congenital CMV disease [69,70,71,72]. 

#### 3.1.9. CMV-Specific T Cells

Virus-specific T cells (VSTs), as a means to adoptively transfer immunity from HSCT donors to recipients, were first tested in the early 1990s [73]. Since that time, a large number of methods for the ex vivo isolation, expansion and enrichment of CMV VSTs have been described [74,75]. The goal of this approach is to reconstitute anti-CMV immunity in patients whose risk of clinically significant infection is due to poor cellular immune function. The majority of the clinical trial experience has been in recipients of HSCT, in whom clinically significant CMV infection can be effectively prevented or controlled with the infusion of CMV VSTs from their transplant donor [76,77,78,79,80,81,82]. A major limitation to the widespread use of CMV VSTs is the bespoke nature of donor-derived product manufacture intended for a single recipient. To this end, off-the-shelf VST products have been developed and have the advantage of advanced manufacture and rapid availability [83,84,85]. Third-party donor-derived VSTs have shown excellent clinical efficacy and safety in phase I and II trials in the salvage setting and when administered early in the course of CMV reactivation. Several randomised trials are underway. The clinical testing of CMV VSTs in non-HSCT patients such as primary immune deficiency or solid transplant recipients is limited but some cases of successful outcomes have been reported [74]. Widespread access to VST therapy remains elusive, largely due to logistic barriers. As manufacturing becomes more straightforward, cell banks become more widely accessible and larger studies with randomised study design report outcomes, it is anticipated that VSTs will become a part of the routine management of CMV infection.

### 3.2. Therapeutic Options for Latent CMV 

CMV latency is a state where the viral genome is maintained in the cell, with the ongoing transcription of a variable number of genes but without production of infectious viral particles [10]. Pro-inflammatory and myeloid differentiation activity has the potential to reactivate these latently infected cells, resulting in the production of infectious CMV virions. 

All the currently available CMV antiviral therapeutics target replication events and are therefore inactive against latent, i.e., non-replicating, virus. Vincristine, which is transported out of cells by multi-drug resistance-associated protein-1 (MRP-1), is downregulated by CMV UL138 expressed during CMV latency, resulting in increased vincristine toxicity in CMV latently infected cells, and vincristine has been demonstrated to kill CMV latently infected cells in vitro [86]. Another approach to killing CMV latently infected cells is using an engineered fusion toxin protein (F49A-FTP). The F49A-FTP binds US28, which is expressed on the surfaces of infected cells, leading to internalisation, where F49A-FTP kills the CMV-infected cells [87]. To the authors’ knowledge, this agent is not yet undergoing human trials.

### 3.3. Cytomegalovirus Vaccines

The development of an effective CMV vaccine would be a significant advance for the management of patients receiving iatrogenic immune suppression. A CMV vaccine would also decrease the rates of CMV infection during pregnancy and in the perinatal period, thus reducing the morbidity of congenital and perinatal CMV. Any CMV vaccine would need to elicit both a strong anybody and cellular response to induce protective immunity. No effective CMV vaccine has been developed to date. A live-attenuated CMV vaccine, derived from the Towne strain, had minimal protection in renal transplant patients or seronegative women [88,89]. More recently, the development of a CMV subunit vaccine based on pp65, an abundant tegmentum protein, and gB, a glycoprotein expressed on the infected cell surface, resulted in boosted antibody, demonstrating up to 50% protection of mothers for CMV infection and recipient-negative/donor-positive solid organ transplant recipients [90,91]. The success of mRNA vaccines in SARS-CoV-2 has stimulated investment in this technology for other pathogens, including CMV. A multi-mRNA vaccine (mRNA-1647) is undergoing dose finding and immunogenicity trials at the current time.

## 4. Conclusions

CMV continues to cause significant morbidity and mortality in children with compromised immune systems. The international screening program using TREC is identifying infants with SCID and other T cell lymphopenic diseases, allowing risk modification to prevent opportunistic infections including CMV [34]. This allows for early curative HSCT, which has been demonstrated have a survival benefit if performed under the age of 3 months [34]. CMV is well documented and reported in the allogeneic HSCT patient population but with the current therapies still being ineffective or with significant side effects, leading to difficult CMV treatment. The data on CMV-directed immunotherapy are promising, with recent reports of randomised clinical trials of CMV-directed immunotherapy.

Given the lack of data on the incidence and treatment of CMV outside HSCT, and with the introduction of new agents including CAR T, this sphere needs urgent further investigation and clarification. It is likely that there are unique groups of susceptible patients receiving CAR T and other immunotherapies that will have significant clinical CMV. For these immunosuppressed children, we advocate the identification of at-risk children and strategies for the prevention of CMV primary infection, particularly in infants with severe primary immunodeficiencies. For children with secondary immunodeficiencies, we suggest CMV reactivation monitoring and CMV treatment to prevent CMV disease. 

## Data Availability

Not applicable.

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
