# Peer review of "Cytomegalovirus Infections in Children with Primary and Secondary Immune Deficiencies"

_viruses, 2021, doi:10.3390/v13102001_

Round 1

Reviewer 1 Report

Bateman and colleagues set out to review the literature on CMV infection in children with primary and secondary immune deficiencies. The authors give an overview on the clinical status-quo and list available therapies or prevention strategies, although I found parts of the review not focused enough on the subject of the title. The article is well-written and cites appropriate literature. Besides some wording and minor concerns regarding structure, this is an interesting read for clinicians and researchers in the field.

Comments:

1) The abstract is very short and should elaborate a bit on the take-home message this review can provide. What does this mean in clinical practice, for diagnostics or for potential prevention? In addition, diagnostics are discussed (TRECs) in the review but are not mentioned as part of the review. In my opinion, diagnostics could also be mentioned in the abstract.

2) The abstract used the phrase “equipped with mechanisms”. This sounds strange and should be written differently. The virus genome rather encodes a variety of immune modulators and evasins.

3) The abstract mentions a focus on reactivation, but the title and the main text do actually not exclude primary infection. I found this misleading.

3) The Introduction is very short and cites only very little literature. It seems to be unclear why this paragraph is needed besides introducing the virus. I would suggest using the introduction paragraph to also give an overview on the current health impact with some numbers as well as some general summary on current diagnostics and treatment/prevention strategies (just the class of molecules and direction of strategies). The final sentence now states a slightly different focus than the final sentence in the abstract. To me, this sounds inconsistent.

5) line 45/46. There is more evidence in this direction and beyond CD34+ cells (https://doi.org/10.1128/mBio.00013-18)

6) lines 46-48. I would appreciate more information about immune evasion (see https://doi.org/10.1038/s41577-019-0225-5) as this is a defining feature of herpesviruses and might play a major role in the lack of HIG efficiency as mentioned in paragraph 3.1.8 (10.7554/eLife.63877, https://doi.org/10.1371/journal.ppat.1004131, 10.3389/fimmu.2019.02110).

7) The list of therapeutic strategies does not adhere to the patient group in the center of this review, immunocompromised children. While I find this list to be well-written and certainly of great use, the authors might want to put these paragraphs more into the context of the rest of the review and add more specific knowledge on sequelae or efficacy in children, if such data exists. In fact, the words “children” or “paediatric” are not found once in the entire paragraph 3.

8) Regarding the role of HIG in cCMV, Kagan et al actually do find an effect in a very recent study (https://doi.org/10.1002/uog.23596). Then this even more recent study again finds no effect (10.1056/NEJMoa1913569) The picture really is not that clear and no clear conclusion can be drawn regarding HIG and cCMV. However, dosage and time of application seem to play a big role. This could be discussed to not give the impression that HIG is useless.

9) A final statement on the currently used strategies for immunocompromised children and possible new strategies in more detail, beyond the one mentioned, would be appreciated. Current strategies, as far I found, are only mentioned briefly and not in detail in lines 225-228.

Reviewer 2 Report

This is a well-written and interesting review that contributes to the field by compiling information about CMV disease in treatment, with a focus on immune compromised children.  The manuscript describes the impact of CMV on children with severe-combined immune deficiency (SCID), leukemia, HLH, and other conditions, and also discusses currently available treatments and therapeutic options.  In particular, the descriptions of establishment of latency in progenitor cells and risk of reactivation are well done. The manuscript is an excellent resource for those new to CMV biology and for more seasoned virologists.

Attention to a few minor items would improve the manuscript overall:

  1. Can you provide some frame of reference for the number of children affected by these immunodeficiencies each year? Either in Australia specifically or worldwide. It would be helpful to know whether there are thousands of kids getting HSCT each year or dozens in order to put this problem in perspective.
  2. Lines 135-136 It is not clear what is meant by the breadth of genetic susceptibility to CMV infection.  Is this adding the NOS2 deficiency to the list that includes SCID and HLH, or are there others? 
  3. The description of the drugs is excellent and very thorough. However, several of the drugs have have significant toxicities or contraindications for use in children.  It is not clear under what circumstances foscarnet or cidofovir would be used in  kids – can you elaborate on this for each of these sections (3.1.3, 3.1.4)? 
  4. Lines 442-444 The reference here is a 2017 paper.  Are there any updates on the status of F49A-FTP, has is advanced to clinical trials, or is this in the works?  Please comment.

Reviewer 3 Report

This is a well written, interesting manuscript, employing appropriate methodology that will contribute to the literature.

Author Response

We thank reviewer 3 for their kind comment.